# Virtual Reality-Based Assessment for Rehabilitation of the Upper Limb in Patients with Parkinson’s Disease: A Pilot Cross-Sectional Study

**DOI:** 10.3390/medicina60040555

**Published:** 2024-03-29

**Authors:** Luciano Bissolotti, Justo Artiles-Sánchez, José Luís Alonso-Pérez, Josué Fernández-Carnero, Vanesa Abuín-Porras, Pierluigi Sinatti, Jorge Hugo Villafañe

**Affiliations:** 1Fondazione Teresa Camplani Casa di Cura Domus Salutis, 25123 Brescia, Italy; luciano.bissolotti@ancelle.it; 2Musculoskeletal Pain and Motor Control Research Group, Faculty of Health Sciences, Universidad Europea de Canarias, C/Inocencio García 1, 38300 Santa Cruz de Tenerife, Spain; jartiles71@gmail.com; 3Department of Physiotherapy, Faculty of Health Sciences, Universidad Europea de Canarias, 38300 Santa Cruz de Tenerife, Spain; 4Department of Physiotherapy, Faculty of Sport Sciences, Universidad Europea de Madrid, 28670 Villaviciosa de Odón, Spain; vanesa.abuin@universidadeuropea.es (V.A.-P.); plgsinatti@gmail.com (P.S.); 5Musculoskeletal Pain and Motor Control Research Group, Faculty of Sport Sciences, Universidad Europea de Madrid, 28670 Villaviciosa de Odón, Spain; 6Onelife Center, Multidisciplinary Pain Treatment Center, 28925 Alcorcón, Spain; 7Department of Physical Therapy, Occupational Therapy, Rehabilitation and Physical Medicine, Rey Juan Carlos University, 28032 Madrid, Spain; josue.fernandez@urjc.es; 8Grupo Multidisciplinar de Investigación y Tratamiento del Dolor, Grupo de Excelencia Investigadora URJC-Banco de Santander, 28922 Madrid, Spain; 9Motion in Brains Research Group, Institute of Neuroscience and Movement Sciences (INCIMOV), Centro Superior de Estudios Universitarios La Salle, Universidad Autonoma de Madrid, 28049 Madrid, Spain; 10La Paz Hospital Institute for Health Research, IdiPAZ, 28029 Madrid, Spain

**Keywords:** stroke, sexual dysfunction, trunk stability

## Abstract

*Background and Objectives*: This study aimed to examine the responsiveness and concurrent validity of a serious game and its correlation between the use of serious games and upper limbs (UL) performance in Parkinson’s Disease (PD) patients. *Materials and Methods*: Twenty-four consecutive upper limbs (14 males, 8 females, age: 55–83 years) of PD patients were assessed. The clinical assessment included: the Box and Block test (BBT), Nine-Hole Peg test (9HPT), and sub-scores of the Unified Parkinson’s Disease Rating-Scale Motor section (UPDRS-M) to assess UL disability. Performance scores obtained in two different tests (Ex. A and Ex. B, respectively, the Trolley test and Mushrooms test) based on leap motion (LM) sensors were used to study the correlations with clinical scores. *Results*: The subjective fatigue experienced during LM tests was measured by the Borg Rating of Perceived Exertion (RPE, 0–10); the BBT and 9HPT showed the highest correlation coefficients with UPDRS-M scores (ICCs: −0.652 and 0.712, *p* < 0.05). Exercise A (Trolley test) correlated with UPDRS-M (ICC: 0.31, *p* < 0.05), but not with the 9HPT and BBT tests (ICCs: −0.447 and 0.390, *p* < 0.05), while Exercise B (Mushroom test) correlated with UPDRS-M (ICC: −0.40, *p* < 0.05), as did these last two tests (ICCs: −0.225 and 0.272, *p* < 0.05). The mean RPE during LM tests was 3.4 ± 3.2. The evaluation of upper limb performance is feasible and does not induce relevant fatigue. *Conclusions*: The analysis of the ICC supports the use of Test B to evaluate UL disability and performance in PD patients, while Test A is mostly correlated with disability. Specifically designed serious games on LM can serve as a method of impairment in the PD population.

## 1. Introduction

Parkinson’s disease (PD) is one of the most common neurodegenerative disorders worldwide [1]. Data have been published about the history of the natural evolution of signs and symptoms in patients with PD [2]. However, this neurodegenerative progression stresses the need to increase the amount of knowledge about the natural course of the disease and the transition model from impairment to clinical complications [3]. Current clinical models elucidate the decline in quality of life in correlation with the Hoehn and Yahr classification progression, the annual rate of disability deterioration (UPDRS), and the diminishing upper limb performance due to disease progression [4,5].

Virtual reality (VR) technology has emerged as a promising tool in the field of rehabilitation, offering a wide range of applications across various medical conditions and therapeutic settings [6]. VR is utilized to enhance traditional physical therapy exercises and activities. Patients can engage in immersive virtual environments that simulate real-life scenarios, such as walking on a virtual beach or climbing virtual stairs, which can aid in restoring mobility, balance, and coordination [7]. It can be used to address cognitive impairments resulting from strokes, traumatic brain injuries, or neurodegenerative disorders. VR environments can be designed to challenge memory, attention, problem-solving, and other cognitive functions, helping patients relearn and improve these skills. VR is being explored for neurorehabilitation in conditions like PD and multiple sclerosis. Virtual reality systems can provide sensory feedback and motor training, aiding in improving motor function, balance, and coordination in these patients. By offering augmented feedback about performance, enabling the individualized repetitive practice of a motor function, and stimulating both motor and cognitive processes simultaneously, VR offers opportunities to learn new motor strategies and to relearn motor abilities that were lost as a result of injury or disease [8]. 

In recent years, various types of exergames based on virtual reality (E-VR) systems, under the Serious Games category, have been introduced in rehabilitation to target upper limb (UL) exercises in PD patients [9,10]. Recently, a newly designed exergame system, named Gloreha ARIA (Idrogenet Srl, Lumezzane, Italy) and based upon a serious game—defined as a game with a primary purpose different from pure entertainment, such as an education game with applications in health, exercise therapy, politics, and security—proposal mediated by leap motion (LM), was introduced in the field of rehabilitation [11]. With the increase in the number of E-VR systems, the importance of understanding the regulatory principles of UL rehabilitation with serious games is relevant for the achievement of good outcomes when treating PD patients in an attempt to slow the progression of UL functional loss [12,13]. 

It is crucial to comprehensively determine and assess the reliability of newly designed E-VR tools for assessing upper limb performance in this population and to meticulously study the correlation and relationship of kinematic scores derived from these devices. This invaluable information empowers clinicians to accurately document meaningful functional changes in upper limb and hand dexterity, ultimately aiding in evaluating the true effectiveness and impact of rehabilitation interventions. 

Therefore, ensuring and guaranteeing the reliability, dependability, and consistency of measurements of tests based on this kind of virtual reality system is indispensable, as it is essential for successful and satisfactory data collection and the reliable interpretation of the results. To date, however, it remains unclear how VR technology may be optimally used and adjusted to the specific needs of various patient populations. A high-quality study is needed to examine the responsiveness and concurrent validity of this new training approach methodology. Specifically, responsiveness and concurrent validity hold significant clinical importance in ensuring the accuracy of follow-up outcomes. The existence of good responsiveness and concurrent validity enables longitudinal comparisons to be conducted over a prolonged duration of time. Dependable and consistent findings enable professionals to arrive at conclusions that are least influenced by extrinsic factors, thus lowering the risk of errors. Therefore, the purpose of this study was to examine the responsiveness and concurrent validity of the kinematic tests proposed by the Gloreha ARIA exergaming system in subjects who suffer from PD.

## 2. Materials and Methods

This study adopted a non-prospective observational cross-sectional design. Twenty-four consecutive community-dwelling patients (14 males, 10 females) who were diagnosed with idiopathic PD by a neurologist, according to the UK Brain Bank Parkinson’s criteria, were included in the study protocol. They were enrolled consecutively during a medical examination in a rehabilitation department. The age of the participants ranged from 63 to 85 years (mean 74.7 ± 6.2 years old, BMI 25.2 ± 4.6 kg/m^2^); none exhibited severe dyskinesias and were not allowed to attend the exergame session. All the patients included in the study were independent in walking, in some cases through the use of assistive devices. The PD group followed their normal medication regimen during testing and the functional evaluation was performed during the ON phase, intended as the following 60–90 min far from the last administration of the last dose of therapy, to discriminate it from the OFF phase (12 h after withdrawal of medication), because during the OFF phase, individuals may experience increased tremors, muscle stiffness, difficulty initiating movements, freezing episodes, and other motor complications. The treatment approach followed for our patients for the initial dosing of carbidopa/levodopa: one tablet of levodopa 25/100 (25 mg of carbidopa and 100 mg of levodopa) three times per day; the dosage was then adjusted gradually based on the individual’s response to the medication. The total daily dose can range from 300 mg to 1200 mg of levodopa. Exclusion criteria included Parkinsonian disorders, such as progressive supranuclear palsy, Shy-Drager syndrome, corticobasal degeneration, secondary parkinsonism, familial Parkinsonism, and a history of dementia as reported by the family or caregiver. In addition, the participant’s ability to follow simple instructions, as determined by their responses to questions and instructions, was assessed during the medical visit and consent process. Informed consent was received from each participant.

### 2.1. Clinical Measurements

Within a multidimensional evaluation model, the Hoehn and Yahr scale (H&Y, range 0–5) was used to define the severity of the disease while the Unified Parkinson’s Disease Rating-Scale Motor section (UPDRS-M) was used to assess disability [14]. Clinical measurements, such as the Box and Block test (BBT) and the Nine-Hole Peg test (9HPT), were used to provide a measure of manual dexterity [15,16,17,18]. The BBT measures unilateral gross manual dexterity. It is composed of a wooden box divided into two compartments by a partition and 150 blocks of 2.5 cm^3^ placed on the side of the partition with the testing hand. The BBT administration consists of asking the patients to move, one by one, the maximum number of blocks from one compartment to another, over a 15.2 cm tall partition, within 60 s. A subject’s score is equal to the number of blocks transported [18]. The subject can select blocks in any order to transport them over the partition as quickly as possible, with the only requirement being that the subject’s fingertips cross the vertical plane of the partition. The 9HPT is used to measure finger dexterity and consists of taking nine pegs from a container, one by one, and placing them into the holes on a board, as quickly as possible, using only the hand of the side being tested. After that, participants have to remove the pegs from the holes, one by one, and replace them in the container. The performance times were determined on the original wooden square version described by Mathiowetz et al. and the time needed to complete the 9HPT in seconds was recorded [18]. The numeric rating scale (NRS), from 0 to 10, was adopted to determine the presence of pain in the upper limbs. The Borg scale [19] was utilized, at the end of each exergame test, to measure the perceived subjective fatigue experienced (RPE) during the exergame. All the functional scores were collected by the same evaluator. All tests were administered to both upper limbs—to acquire data for the comparison of the sides—and the total time of the test session was 40 min.

### 2.2. Measurements during the Exergaming Test

The Gloreha ARIA (Idrogenet Srl, Lumezzane, Italy) exergaming system was used in this study. This exergaming system aims to offer cognitive exercises and interactive games focusing on free arm, wrist, and hand movements. This occurs throughout several challenging and recreational exercises based on active upper limb movements detected by a specific sensor (LM controller), an optical hand-tracking module that captures the movements of the hands with a good accuracy level. The exercise difficulty level can be programmed by the therapist or can auto-adjust based on the patient’s performance. These games have been created to engage and increase the active participation of the subject in the rehabilitation program. The ARIA system detected the range of movement of finger flexion–extension, wrist pronation–supination, radial–ulnar deviation, wrist flexion–extension, and arm movements in vertical and horizontal planes. The exergaming system runs on a desktop and enables the clinician to deliver the rehabilitation session by exploiting the LM sensor. Two different types of exergames were presented to the patients. During the execution of the game, the patient maintained the sitting position, and the upper limb weight was partially relieved by a dynamic arm support (DAS) system that performed the calibration operation over the LM sensor. The DAS supported the elbow and the proximal portion of the forearm, to assist the upper limb movements by using a compensatory mechanism that allowed the patients to move with reduced gravity.

#### 2.2.1. Exercise A (Ex. A): The Trolley Test

This game consisted of collecting objects falling from the ceiling in a trolley to collect the maximum number possible of objects within a fixed amount of time (40 min). During the game, the participants needed to move the tested upper limb and the corresponding hand in horizontal movements in abduction and adduction of the shoulder to collect the falling objects. The objects’ shapes recall daily life common objects or tools, like fruits, foods, and/or boxes. Series were played to obtain a balanced distribution of falling objects on both sides of the screen. The number of falling objects and the fall speed of these were maintained stable for all the patients. During the performance of the game, arm posture control was required, keeping the hand over the LM device that virtually enabled the trolley to collect objects. The number of collected objects at the end of the test was used as a performance score.

#### 2.2.2. Exercise B (Ex. B): The Mushroom Test

This game consisted of reaching and grasping a mushroom that appeared on a screen at different depths in the virtual environment. As the mushroom is reached, the patient has to grab it from the grass and then bring and release it beyond a line close to the patient. The mushrooms on the screen are located at different heights and depths, stimulating the patients’ spatial perception and making them move their arms in the space above the LM sensor until the correct position of the target mushroom is found. During this game, the patient’s virtual hand was asked to perform forward and backward movements on the sagittal plane, motivating the users to move the upper limb away from the body to reach the virtual mushroom, thus making specific movements of the flexion–extension of the elbow, hand, and fingers to reach and bring the virtual object. During this game, the subject trained their gross and fine motor unilateral coordination in reaching, grasping, and releasing objects. Five patients were involved in the analysis of concurrent validity, and they were invited to repeat the two trials at another time (the second time was 24 h later than the first one). The examiner ensured that the patients were in the ON phase when they performed the exercises and that the drug therapy was not changed in timing and dosages at the second trial time.

### 2.3. Statistics

The data are depicted using the SPSS 28.0 software (SPSS Inc., Chicago, IL, USA). Descriptive statistics (mean and standard deviation) were provided for all patients. Pearson correlation coefficients are commonly used to assess the relationship between two continuous variables. The relationships between each functional parameter and age, years from diagnosis, H&Y, and UPDRS-M were assessed using Pearson correlation coefficients. The Pearson correlation coefficient measures the strength and direction of the linear relationship between two variables, ranging from −1 to 1. Statistically significant correlation coefficients were considered clinically large if above 0.5, moderate between 0.49 and 0.3, and small between 0.29 and 0.1 [18]. A level of significance at 0.05 was used for the statistical analyses. A *t*-test was used to evaluate differences between the right and left upper limbs.

## 3. Results

### 3.1. Clinical Characteristics of the Participants

At the time of observation, the PD patients were staged as belonging to classes 1–2 in 43% of the cases, 2.5–3 in 24%, and 4 in 33%, according to Hoehn and Yahr classification. Years from diagnosis were 6.5 years ± 4.3; range: 1.5–17. Male patients were the majority and they were all right-hand dominant. According to subjective opinion, 32% presented the worst impairment on the right side and 55% on the left one, while the remainder were not able to identify it precisely. UL disability and impairment were not statistically different upon side-to-side comparison when assessed with UPDRS-M, BBT, 9HPT, or exergaming tests performed with Gloreha Aria. The RPE (1–10) revealed an overall moderate perceived fatigue while performing exergaming tests, with no difference from right to left UL.

### 3.2. Analysis of Correlations

The analysis of concurrent validity showed a large correlation coefficient for Ex. B (0.927, *p* < 0.001), Ex. A (0.791, *p* = 0.006), 9HPT (0.773, *p* = 0.009), and BBT (0.818, *p* = 0.004). The duration of the disease (years from diagnosis) was not correlated with the performance of the right and left upper limbs, according to UPDRS-M, BBT, 9HPT, or exergames scores. UPDRS-M scores presented a negative correlation with the BBT test, Ex. A, and Ex. B (*p* < 0.05), while it was positively correlated to 9HPT’s value (*p* < 0.05) (Table 1). BBT and 9HPT correlated with each other (*p* < 0.05).

## 4. Discussion

VR tools have gained significant attention in the research focused on assessing and providing rehabilitation treatment for the upper limb in people affected by neurological diseases. These tools offer a novel approach to rehabilitation by creating immersive and interactive environments that can enhance engagement, motivation, and motor learning. The current research presents an analysis of VR tools created to assess and provide rehabilitation treatment of the upper limb in people affected by neurological diseases. Special attention was given to PD patients to describe a standardized tool to assess and treat UL impairment in this population of patients. The present study aimed to investigate the concurrent validity of the kinematic tests proposed by the Gloreha ARIA exergaming system in subjects with PD. To guarantee a multidimensional evaluation model, a battery of tests (H&Y for severity of disease, UPDRS-M, 9HPT, and BBT) was conducted in addition to the exergaming system’s exercises, and its concurrent validity showed a large correlation coefficient in 9HPT and BBT tests as well in both exergaming exercises (Ex. A and Ex. B). 

Upon careful observation and consideration of the current neurologic treatment options, it is evident that VR-based rehabilitation is rapidly expanding and evolving. This innovative approach has emerged as an effective and powerful therapeutic tool, facilitating motor learning and offering the potential to significantly improve balance and gait in neurologic patients. Furthermore, when combined with conventional rehabilitation, VR-based rehabilitation can provide additional benefits and enhance the overall treatment experience and outcomes [20]. However, the evidence has not yet resulted in standardized guidelines, probably motivated by insufficient methodological quality and the lack of randomized clinical trials with robust methodological designs [21]. In this context, the development of economic evaluations in the area of virtual reality applied to the rehabilitation of patients with Parkinson’s is crucial, as is occurring in other medical areas where virtual reality is being applied [22,23,24,25,26], to identify its cost-effective relationship with other relevant diagnostic and treatment alternatives in terms of resource consumption and effectiveness [27,28]. Incremental cost-effectiveness analyses can be very useful for evaluating the application of virtual reality in the rehabilitation of Parkinson’s patients, determining whether the additional outcomes obtained (improvement of motor function, quality of life, functional independence, etc.) by incorporating virtual reality justify the additional costs associated with this technology compared to conventional rehabilitation [29]. Cost-effectiveness analyses provide valuable information for the efficient allocation of resources in the healthcare system. In this case, if virtual reality is demonstrated to be a cost-effective option for the rehabilitation of Parkinson’s patients, healthcare systems can strategically plan and invest in this technology to maximize its impact on affected population health. These evaluations can guide decision making aimed at the adoption, implementation, and funding of virtual reality technologies in patients affected by Parkinson’s. 

The primary outcome of our study demonstrated that the Gloreha ARIA system can be confidently regarded as a dependable and trustworthy tool for evaluating upper limb impairment and disability in patients with PD. This VR system possesses the potential to provide a consistent and reliable environment for patients, clinicians, and researchers to engage in accurate and reproducible motor and cognitive rehabilitation. As a result, this technology could facilitate and advance our ability to effectively diagnose, treat, and manage PD-related upper limb impairments and disabilities. Furthermore, in a multidimensional disability, such as PD, VR tools create the condition to ease access to rehabilitation care for many patients with upper limb motor or sensory impairments [30]. In the population of patients being studied, it is important to highlight the results obtained from the assessment. The correlations observed between VR exercises and functional scores provide valuable insights for clinicians. These findings indicate that VR devices can be used as a measurement system to assess the progression of a disease or functional improvement during the treatment phase. Using this VR tool, the clinician can introduce the patient to research or rehabilitation protocols without imposing excessive fatigue on the subject as shown by the low values of rate of perceived exertion measured by Borg at the end of the sessions. In the side-to-side comparison analysis, the results we obtained to date do not show consistent differences either according to the usual metric scores or VR system. These data support the reliability of this VR device in evaluating UL performance in PD patients as it is possible to make in a “real-world” setting. The type of tasks imposed by Ex. A is useful to assess the upper limb’s gross motor function, as it demands only the coordinated horizontal abduction/adduction movements of the shoulder and wrist. This exercise presents many similarities to what was asked by the UPDRS section for upper limb assessment and even more by BBT and 9HPT. Either UPDRS and BBT request the repetition of UL movements in elevation and abduction/adduction movements to complete the required motor task, while 9HPT requires any wrist and hand radial and ulnar deviations to complete the test [31,32]. This can explain the wider correlation of Ex. A with all these three clinometric scores. This type of exercise mainly involved the recruitment of the extrinsic muscles of the upper limb, while Exercise B also promoted the involvement of the intrinsic muscles of the forearm and hand. This type of exercise can enhance the fine control of the hand in grasping and releasing objects, while the shoulder and elbow muscles are involved in reaching movements. The motor task of Ex. A imposes a specific training of spatial vision on depth, while Ex. B mostly exercises vision on the horizontal plane. The combination of motor and visual tasks could be useful to enhance the oculomotor coordination in PD patients [33]. This can be of relevant interest in this type of patient, given the impairment of binocular coordination during tests. Since PD imposes a multidimensional rehabilitation strategy, the use of technology can be helpful to satisfy assessment and treatment aims at the same time [34]. Additionally, the correlations observed between VR exercises and functional scores suggest that improvements in VR performance may be indicative of functional improvements in real-world activities. This provides clinicians with a reliable and objective tool to measure the effectiveness of interventions and track patients’ functional progress. Overall, the use of VR devices as a measurement system for assessing the progression of a disease or functional improvement in patients offers promising opportunities for clinicians. The correlations identified between VR exercises and functional scores enhance the clinician’s ability to monitor and evaluate the patient’s progress throughout the treatment phase.

### 4.1. The Clinical Implications

The study highlights the potential of VR tools in rehabilitating upper limb impairments in PD patients. VR technology offers an immersive platform for active engagement and motor learning, showing promise in improving rehabilitation outcomes.

Specifically, VR-based assessments using platforms like the Gloreha ARIA system can effectively evaluate and treat upper limb impairments in PD patients, offering reliable results. VR environments can be customized to address neurological challenges and enhance patient motivation and adherence to treatment.

Integrating VR technology into clinical practice could expand treatment options and improve outcomes for PD patients by enabling tailored interventions and better monitoring of progress. However, limitations, such as the exclusion of certain PD subtypes and lack of a cognitive function assessment, need consideration in future research to fully explore the benefits of VR-based rehabilitation.

### 4.2. Limits

Despite the insights gained from this study, several limitations warrant acknowledgment. Firstly, the exclusion of patients with Parkinsonian disorders other than idiopathic PD, such as progressive supranuclear palsy or corticobasal degeneration, might have introduced selection bias, potentially restricting the applicability of the results to a broader range of Parkinsonian disorders. Additionally, while various clinical measurements were utilized, cognitive function was not specifically assessed. This omission is significant as cognitive impairment is a crucial aspect of PD progression that can influence rehabilitation outcomes.

Furthermore, the absence of a control group limits the ability to compare the effectiveness of VR-based rehabilitation interventions against standard care or other rehabilitation approaches, potentially confounding the interpretation of the results. Moreover, while the study focused on evaluating the concurrent validity of kinematic tests proposed by the Gloreha ARIA exergaming system, the findings may not apply to other VR systems or technologies, thus limiting their generalizability.

## 5. Conclusions

The study underscores the promising potential of VR assessments in addressing upper limb impairments among PD patients. VR technology offers an immersive environment conducive to active participation and motor learning. The results indicate that VR tools, especially specialized serious games on platforms, like LM, are pertinent in neurological rehabilitation for assessing and managing upper limb impairments in PD patients. Furthermore, the study validates the feasibility of employing VR for evaluating upper limb performance without inducing notable fatigue. This emphasizes the adaptability of VR technology in addressing the distinctive challenges associated with neurological diseases.

## Figures and Tables

**Table 1 medicina-60-00555-t001:** Analysis of correlation among exergames scores and clinical tests.

	UPDRS-M (UL)	Box and Block Test (BBT)	Nine-Hole Peg Test (9HPT)	Ex. A	Ex. B
BBT	−0.392 *	-	−0.622 *	0.390 *	0.272
9HPT	0.557 *	−0.622 *	-	−0.447 *	−0.225
Ex. A	−0.392 *	0.390 *	−0.447 *	-	-
Ex. B	−0.325 *	0.272	−0.225	-	-

*: *p* < 0.005.

## Data Availability

Data are contained within the article.

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
