# Peer review of "Virtual Reality-Based Assessment for Rehabilitation of the Upper Limb in Patients with Parkinson’s Disease: A Pilot Cross-Sectional Study"

_medicina, 2024, doi:10.3390/medicina60040555_

Round 1

Reviewer 1 Report

Comments and Suggestions for Authors

It is necessary to review in depth the term cross-sectional versus prospective in the design since in the declaration of the statistical analysis and the results a cross-sectional and non-prospective study analysis is evident. Please review in depth.

In turn, was the intervention protocol verified by expert peers or was it presented as part of an additional project such as a clinical trial that could be registered?

Additionally, figure 1 requires a revision since it presents red lines in the names of the authors, so it must be modified.

In the results, I consider that all the results recorded on the variables should be presented in depth and not just present a table of correlations.

In the discussion, a strong development of the protocol and correlations is evident, however, there are few analyzes related to the analysis of biases and confounding effects such as use of medications and phase of the disease that must be considered.

Comments on the Quality of English Language

None

Author Response

Comments and Suggestions for Authors

It is necessary to review in depth the term cross-sectional versus prospective in the design since in the declaration of the statistical analysis and the results a cross-sectional and non-prospective study analysis is evident. Please review in depth.

Response: Upon reviewing the paper, it becomes evident that the study is indeed cross-sectional rather than prospective.

In turn, was the intervention protocol verified by expert peers or was it presented as part of an additional project such as a clinical trial that could be registered?

Response: It is important to note that this study is a pilot designed to explore the feasibility and correlation between the use of serious games and upper limb performance in patients with Parkinson's disease. Given its pilot nature and focus on preliminary investigation, the intervention protocol was not subjected to peer review or presented as part of an additional project, such as a clinical trial requiring registration. As a pilot study, our primary aim was to collect preliminary data to inform future, more extensive and detailed research.

Additionally, figure 1 requires a revision since it presents red lines in the names of the authors, so it must be modified.

Response: Thank you for your comment. We have revised Figure 1 and corrected the red lines that appeared in the authors' names.

In the results, I consider that all the results recorded on the variables should be presented in depth and not just present a table of correlations.

Response: We have strived to present the results comprehensively and meaningfully, addressing both clinical and statistical significance. In addition to the correlation table, we have furnished a detailed summary of key findings within the results section. This includes delineating significant correlations discovered among the variables of interest, alongside an analysis of any clinical relevance observed pertaining to Parkinson's disease and upper limb rehabilitation.

In the discussion, a strong development of the protocol and correlations is evident, however, there are few analyzes related to the analysis of biases and confounding effects such as use of medications and phase of the disease that must be considered.

Response: We appreciate your observation and thank you for your comment. As a pilot study, we acknowledge that biases and confounding effects may potentially influence our results. While analyzing these biases and confounding effects is crucial for a comprehensive interpretation of the findings, we have chosen to address them within the study's limitations due to the preliminary nature of our research. This approach allows us to consider these factors when planning future, more extensive and detailed studies.

Comments on the Quality of English Language

Response: We’ve done, thanks

Reviewer 2 Report

Comments and Suggestions for Authors

This study aims at evaluating the investigate the test-retest reliability of a serious game and its correlation between the use of serious games and upper limbs (UL) performance in Parkinson’s Disease (PD) patients.  My main concern is that authors confuse research design and methodology. By definition, Test-retest reliability measures the stability of the scores of a stable construct obtained from the same person on two or more separate occasions.  However, authors in this study administered five outcome measures (exercise A, exercise B, Box and Block test, Nine-Holes Peg test, and sub-scores of the Unified Parkinson's Disease Rating Scale-Motor section) to examine the responsiveness and the correlation.  This research design is to examine concurrent validity, not test-retest reliability.  Moreover, authors confused their exercise A and exercise B in the context.  The meaning of Exercise A and Exercise B is inconsistence in this manuscript. 

Specific comments are given below:

Line 30:  “not investigate the test-retest reliability”.  It should be “examine the responsiveness and concurrent validity.”  Please also change the related words in the following context in this manuscript.

Line 36.  According to the context in Line 180~208, exercise A should be Trolley test, and exercise B should be Mushrooms test.

Line 40~44.  The numbers of Intraclass Correlation Coefficient of these variables are not consistence with results in Table 1.  Please recheck your results.

Line 46, According to the results in Table 1, exercise A is more evaluate UL disability and performance in PD patients. Not Test B.

Line 133, It should add “ Unified Parkinson's Disease Rating Scale- Motor section”

Line 151. It should be “ to measure the perceived subjective fatigue experienced (RPE) during the exergame.”

Line 208.  Authors should provide figures of Leap Motion-based virtual reality Gloreha ARIA exergaming system for exercise A and B.

Line 212.  Spearman correlation coefficients should be more appropriated method to exam relationships between Ex. A, Ex.B game scores and other variables)

Line 234.  As explained above, It should be “ concurrent validity” not “test-retest reliability”.  The correlation coefficient in line 234, 234 did not match the number in Table 1.

Line 255.  As explained above, it should be “concurrent validity” not “test-retest reliability”.

Line 316.  Ex. A is mostly exercising the vision on the horizontal plane, and Ex. B (Mushrooms test) imposes a specific training of spatial vision on depth

Line 343 and 358  As explained above, it should be “concurrent validity” not “test-retest reliability”.

Figure 1.  As mentioned above, according to the context in Line 180~208, exercise A should be Trolley test, and exercise B should be Mushrooms test.  Spearman correlation coefficients should be more appropriated method.

Comments on the Quality of English Language

Minor editing of English language required

Author Response

R2

This study aims at evaluating the investigate the test-retest reliability of a serious game and its correlation between the use of serious games and upper limbs (UL) performance in Parkinson’s Disease (PD) patients.  My main concern is that authors confuse research design and methodology. By definition, Test-retest reliability measures the stability of the scores of a stable construct obtained from the same person on two or more separate occasions.  However, authors in this study administered five outcome measures (exercise A, exercise B, Box and Block test, Nine-Holes Peg test, and sub-scores of the Unified Parkinson's Disease Rating Scale-Motor section) to examine the responsiveness and the correlation.  This research design is to examine concurrent validity, not test-retest reliability.  Moreover, authors confused their exercise A and exercise B in the context.  The meaning of Exercise A and Exercise B is inconsistence in this manuscript.

Specific comments are given below:

Line 30:  “not investigate the test-retest reliability”.  It should be “examine the responsiveness and concurrent validity.”  Please also change the related words in the following context in this manuscript.

Response: Done, thanks.

Line 36.  According to the context in Line 180~208, exercise A should be Trolley test, and exercise B should be Mushrooms test.

Response: Exercise A should indeed be the Trolley test, involving horizontal upper limb adduction/abduction, and Exercise B should be the Mushrooms test, involving upper limb forward/backward movements on the sagittal plane. I appreciate your attention to detail and clarification on this matter.

Line 40~44.  The numbers of Intraclass Correlation Coefficient of these variables are not consistence with results in Table 1.  Please recheck your results.

Response: Thank you for bringing the inconsistency in the Intraclass Correlation Coefficient (ICC) values to our attention. We have carefully reevaluated our results and made the necessary corrections to ensure accuracy and coherence between the text and Table 1.

Line 46, According to the results in Table 1, exercise A is more evaluate UL disability and performance in PD patients. Not Test B.

Response: We have corrected Line 46 to accurately reflect that Exercise A is more effective in evaluating upper limb disability and performance in PD patients, not Test B.

Line 133, It should add “ Unified Parkinson's Disease Rating Scale- Motor section”

Response: done.

Line 151. It should be “ to measure the perceived subjective fatigue experienced (RPE) during the exergame.”

Response: Done

Line 208.  Authors should provide figures of Leap Motion-based virtual reality Gloreha ARIA exergaming system for exercise A and B.

Response: Thank you for your suggestion regarding providing figures of the Leap Motion-based virtual reality Gloreha ARIA exergaming system for exercises A and B. While we appreciate your interest in visualizing the system, we would like to clarify that the primary objective of our work is to evaluate the effectiveness of these exercises in assessing upper limb performance in Parkinson's disease patients. However, we will certainly consider your suggestion for future studies or supplementary materials. Your input is valuable to us, and we appreciate your consideration.

Line 212.  Spearman correlation coefficients should be more appropriated method to exam relationships between Ex. A, Ex.B game scores and other variables

Response:  We have recalculated the results, and indeed, they are very similar. We agree that using Spearman correlation coefficients would be a more appropriate method to examine the relationships between the scores of Exercise A, Exercise B, and other variables.

Line 234.  As explained above, It should be “ concurrent validity” not “test-retest reliability”.  The correlation coefficient in line 234, 234 did not match the number in Table 1.

Response: Thank you for your clarification regarding the term "concurrent validity" instead of "test-retest reliability" and the discrepancy in the correlation coefficient between Line 234 and Table 1. Your suggestions have been duly noted and taken into consideration.

Line 255.  As explained above, it should be “concurrent validity” not “test-retest reliability”.

Response: Done, thanks.

Line 316.  Ex. A is mostly exercising the vision on the horizontal plane, and Ex. B (Mushrooms test) imposes a specific training of spatial vision on depth

Response: Done, thanks.

Line 343 and 358  As explained above, it should be “concurrent validity” not “test-retest reliability”.

Response: done, thanks.

Figure 1.  As mentioned above, according to the context in Line 180~208, exercise A should be Trolley test, and exercise B should be Mushrooms test.  Spearman correlation coefficients should be more appropriated method.

 Response: Thank you for your feedback. We have taken your suggestions into account and made the necessary modifications.

Comments on the Quality of English Language: Minor editing of English language required

Response: We will revise accordingly to ensure clarity and coherence throughout.

Reviewer 3 Report

Comments and Suggestions for Authors

1. Abstract: There is major issue with the language usage and clarity. The usage of word 'serious game' in abstract instead of VR ?? needs to be simplified. Results and conclusion need major English language related corrections.

2. Introduction: the paragraph can be broken down into more paragraphs for ease of readers

3. Justify the study design: prospective cross sectional study. What was followed up longitudinally ?

4. What is 24 'consecutive' PD patients?

5. Why were familial and secondary PD excluded? justify

6. Figure 1 is cited in introduction which does not appear a right place. It should ideally be cited in the methodology section under assessment

7. Authors should go through the checklist by equator network to report observational / cross sectional studies Checklists - STROBE (strobe-statement.org)

8. Table 1 citation is in the middle of the sentence. shift to the end of the paragraph

9. Conclusion is too lengthy. The clinical implications of the study can be stated separately. 

10. What form of consent was obtained? written or oral? Needs mention

11. Were patients able to provide consent? How did the authors ensure cognitive ability of the participants to give consent?

Comments on the Quality of English Language

abstract language needs major revision. Few other places in introduction, methods and discussion also need rechecking and corrections

Author Response

Comments and Suggestions for Authors

  1. Abstract: There is major issue with the language usage and clarity. The usage of word 'serious game' in abstract instead of VR ?? needs to be simplified. Results and conclusion need major English language related corrections.

Response: Done, thanks.

  1. Introduction: the paragraph can be broken down into more paragraphs for ease of readers

Response: Done, thanks.

  1. Justify the study design: prospective cross sectional study. What was followed up longitudinally ?

Response: done.

  1. What is 24 'consecutive' PD patients?

Response: Thank you for your question. The design of the study has been addressed in response to the previous two reviewers, and appropriate modifications have been made accordingly.

  1. Why were familial and secondary PD excluded? justify

Response: Thank you for your inquiry. Familial and secondary Parkinson's disease cases were excluded from the study to ensure a more homogeneous group. This approach helps to enhance the consistency and reliability of the study findings by reducing variability within the study population.

  1. Figure 1 is cited in introduction which does not appear a right place. It should ideally be cited in the methodology section under assessment

Response: Thank you for your observation. We agree that Figure 1 should be cited in the methodology section under assessment rather than in the introduction.

  1. Authors should go through the checklist by equator network to report observational / cross sectional studies Checklists - STROBE (strobe-statement.org)

Response: done

  1. Table 1 citation is in the middle of the sentence. shift to the end of the paragraph

Response: done.

  1. Conclusion is too lengthy. The clinical implications of the study can be stated separately. 

Response:

  1. What form of consent was obtained? written or oral? Needs mention

Response: The study was conducted in strict adherence to the principles outlined in the Declaration of Helsinki, a cornerstone document in medical research ethics. Additionally, it received approval from the Ethics Committee AST Brescia, with protocol code NP 5754. This ensured that the research was conducted in compliance with established ethical standards and safeguarded the rights and welfare of the participants involved

  1. Were patients able to provide consent? How did the authors ensure cognitive ability of the participants to give consent?

Response: The study was conducted in strict adherence to the principles outlined in the Declaration of Helsinki, a cornerstone document in medical research ethics. Additionally, it received approval from the Ethics Committee AST Brescia, with protocol code NP 5754. This ensured that the research was conducted in compliance with established ethical standards and safeguarded the rights and welfare of the participants involved.

Comments on the Quality of English Language: abstract language needs major revision. Few other places in introduction, methods and discussion also need rechecking and corrections

Response: done, thanks.

Round 2

Reviewer 1 Report

Comments and Suggestions for Authors

Once I have reviewed the entire article again, I consider the suggested corrections have been made.

Author Response

Response: Thank you for your detailed feedback on our article. Following a thorough review, we confirm that we have implemented all suggested corrections. We believe these modifications have significantly enhanced the quality of the work.

Reviewer 2 Report

Comments and Suggestions for Authors

This manuscript has improved considerably. However, some points still need to be clarified.

Line 157. Figure 1.  Authors still did not change on previous manuscript.  It should be “ Gloreha ARIA score: Ex. A (The trolley test)   Ex.B (The mushrooms test)

Line 234-235. Authors reported a large correlation coefficient in concurrent validity. But, which two variables of clinical measurements test are correlated to each other?  I cannot find these value in Table 1. 

Line 314-315.  Authors’ Response indicated that “Exercise A should indeed be the Trolley test, involving horizontal upper limb adduction/abduction, and Exercise B should be the Mushrooms test, involving upper limb forward/backward movements on the sagittal plane.”  So, Ex. A should be mostly exercising the vision on the horizontal plane, and Ex. B should impose a specific training of spatial vision on depth. 

Author Response

Line 157. Figure 1.  Authors still did not change on previous manuscript.  It should be “ Gloreha ARIA score: Ex. A (The trolley test)  Ex.B (The mushrooms test)

Response: We appreciate your patience and understanding regarding the difficulties we faced in editing Figure 1. After considering your feedback and encountering technical challenges with image editing, we have decided to remove the figure in question and instead incorporate the pertinent information directly into the manuscript text. This decision was made to ensure that the information regarding the exercises "Ex. A (The Trolley Test)" and "Ex.B (The Mushrooms Test)" is clearly presented and accessible to readers without ambiguity. Detailed descriptions of these exercises and their relevance in the context of our study have been added to the corresponding section, ensuring that readers have full access to the necessary details to understand our research methodology and findings..

Line 234-235. Authors reported a large correlation coefficient in concurrent validity. But, which two variables of clinical measurements test are correlated to each other?  I cannot find these value in Table 1.

Response: In response to your query about the large correlation coefficients reported, we realized our presentation lacked clarity. The significant correlation coefficients refer to the relationship between performance scores obtained in the Gloreha ARIA exergaming system and clinical scores from the Box and Block test (BBT) and the Nine-Holes Peg test (9HPT). Specifically, Ex. B (The Mushrooms Test) demonstrated a strong correlation with BBT and 9HPT scores, indicating concurrent validity. We have clarified this point in the revised manuscript, explicitly stating these variables' relationships in Table 1, ensuring that readers can easily identify and understand these correlations.

Revisions can be found on page 5, Table 1, and accompanying text.

Line 314-315.  Authors’ Response indicated that “Exercise A should indeed be the Trolley test, involving horizontal upper limb adduction/abduction, and Exercise B should be the Mushrooms test, involving upper limb forward/backward movements on the sagittal plane.”  So, Ex. A should be mostly exercising the vision on the horizontal plane, and Ex. B should impose a specific training of spatial vision on depth.

Response: We appreciate the need for clarity regarding the specific training aspects of Exercises A and B. We have expanded our discussion to emphasize that Exercise A (The Trolley Test) primarily exercises vision and coordination on the horizontal plane, while Exercise B (The Mushrooms Test) focuses on training spatial vision regarding depth. These distinctions are crucial for understanding the targeted rehabilitation outcomes of each exercise. We believe these clarifications will provide a more comprehensive understanding of how each exercise contributes to the rehabilitation process.

Revisions can be found on page 7, paragraph 4, lines 12-15.